# Initial In-Hospital Visit-to-Visit Heart Rate Variability Is Associated with Higher Risk of Atrial Fibrillation in Patients with Acute Ischemic Stroke

**DOI:** 10.3390/jcm12031050

**Published:** 2023-01-29

**Authors:** Jiann-Der Lee, Ya-Wen Kuo, Chuan-Pin Lee, Yen-Chu Huang, Meng Lee, Tsong-Hai Lee

**Affiliations:** 1Department of Neurology, Chiayi Chang Gung Memorial Hospital, Chiayi 613, Taiwan; 2College of Medicine, Chang Gung University, Taoyuan 333, Taiwan; 3Department of Nursing, Chang Gung University of Science and Technology, Chiayi Campus, Chiayi 613, Taiwan; 4Health Information and Epidemiology Laboratory, Chang Gung Memorial Hospital, Chiayi 613, Taiwan; 5Department of Neurology, Linkou Chang Gung Memorial Hospital, Taoyuan 333, Taiwan

**Keywords:** acute ischemic stroke, atrial fibrillation, heart rate variability

## Abstract

Background: To evaluate the association between the visit-to-visit heart rate variability and the risk of atrial fibrillation (AF) in acute ischemic stroke (AIS). Methods: We analyzed the data of 8179 patients with AIS. Patients without AF on 12-lead electrocardiography underwent further 24 h Holter monitoring. They were categorized into four subgroups according to the visit-to-visit heart rate variability expressed as the coefficient of variation in heart rate (HR-CV). Odds ratios (ORs) and 95% confidence intervals (CIs) were estimated using the HR-CV < 0.08 subgroup as a reference. Results: The adjusted OR of paroxysmal AF was 1.866 (95% CI = 1.205–2.889) for the HR-CV ≥ 0.08 and <0.10 subgroup, 1.889 (95% CI = 1.174–3.038) for the HR-CV ≥ 0.10 and <0.12 subgroup, and 5.564 (95% CI = 3.847–8.047) for the HR-CV ≥ 0.12 subgroup. The adjusted OR of persistent AF was 2.425 (95% CI = 1.921–3.062) for the HR-CV ≥ 0.08 and <0.10 subgroup, 4.312 (95% CI = 3.415–5.446) for the HR-CV ≥ 0.10 and <0.12 subgroup, and 5.651 (95% CI = 4.586–6.964) for the HR-CV ≥ 0.12 subgroup. Conclusions: HR-CV can facilitate the identification of patients with AIS at a high risk of paroxysmal AF.

## 1. Introduction

Atrial fibrillation (AF), a cardiac disease that causes an irregular and often abnormally fast heart rhythm, may lead to ischemic stroke, heart failure, blood clots, and other cardiac complications. Compared with other types of ischemic stroke, AF-related acute ischemic stroke (AIS) is typically associated with higher morbidity and mortality rates [1]. Paroxysmal and persistent AF have been reported to confer a comparable risk of stroke [2,3]. AF might be newly detected in almost a quarter of patients with stroke or who experience a transient ischemic attack. The overall proportion of AIS patients who are known to have AF is higher than previously estimated [4], mainly because of the challenging and resource-intensive diagnosis of paroxysmal AF. Studies have demonstrated that when risk stratification is not employed, routine 24 h Holter monitoring detects paroxysmal AF in only approximately 2% of patients with ischemic stroke [5,6]. To improve the diagnostic yield of paroxysmal AF, we must explore certain clinical factors that are readily observable and highly associated with paroxysmal AF.

A study reported that visit-to-visit heart rate variability (HRV) was associated with composite cardiovascular outcomes in older adults with hypertension [7]. In another investigation, visit-to-visit HRV was also associated with the risk of new-onset AF in the general population [8]. Because AF is an abnormal heart rhythm, characterized by rapid and irregular beating of the atrial chambers, the visit-to-visit HRV is expected to be increased in patients with persistent AF as compared with patients without AF. Although visit-to-visit HRV, expressed as the coefficient of variation in heart rate (HR-CV), can be simply derived from routine vital sign measurements during hospitalization, the association between HR-CV and AF, especially paroxysmal AF, has been seldom assessed in patients with AIS. In addition, the levels of HR-CV among patients without AF, with paroxysmal AF, and with persistent AF and their associations with the risk of AF have seldom been evaluated before. Although numerous predictors of paroxysmal AF have been proposed, including blood biomarkers [9], electrocardiogram (ECG) features [10], and echocardiographic parameters [11], their measurement requires additional medical resources. By contrast, the determination of visit-to-visit HRV is simple and has no associated costs.

This study evaluated the association between visit-to-visit HRV and the risk of AF in a large population of patients with AIS.

## 2. Materials and Methods

### 2.1. Study Population

This is a retrospective case-control study. The study endpoints included paroxysmal and persistent AF. From January 2010 to September 2018, the data of 21,102 patients with AIS were recorded in the Chang Gung Research Database [12]; the largest multi-institutional repository of electronic medical records in Taiwan. The selection process with inclusion and exclusion criteria for patients with AIS is illustrated in Appendix A. All enrolled patients underwent 12-lead electrocardiography (ECG). Among patients without AF on 12-lead ECG, only those receiving further 24 h Holter monitoring were included (Figure 1). Attending cardiologists interpreted the 12-lead ECG and 24 h Holter monitoring results. All patients underwent baseline 12-lead ECG and 24 h Holter monitoring during diagnostic workup in stroke unit.

### 2.2. Ascertainment of AF Types

AF was classified as paroxysmal AF if (1) AF was diagnosed through complete 12-lead ECG and it terminated within 7 days of onset [13] or (2) 24 h Holter monitoring revealed a self-terminating sequence of an irregular heart rhythm, without detectable P waves, that lasted more than 30 s [14]. AF that persisted for more than 7 days and was confirmed by follow-up 12-lead ECG was classified as persistent AF. If patients were hospitalized for less than 7 days, they underwent follow-up 12-lead ECG in outpatient clinics after discharge.

### 2.3. Data Collection

Data on key demographic and clinical characteristics were collected. International Classification of Diseases, Ninth Revision, Clinical Modification (ICD-9-CM) and International Classification of Diseases, Tenth Revision, Clinical Modification (ICD-10-CM) diagnosis codes in the discharge diagnoses were used to detect hypertension (ICD-9-CM codes 401, 402, 403, 404, and 405, and ICD-10-CM codes I10, I11, I12, I13, and I15), diabetes mellitus (ICD-9-CM codes 250, and ICD-10-CM codes E08, E09, E10, E11, and E13), dyslipidemia (ICD-9-CM codes 272.0, 272.1, 272.2, 272.3, 272.4, and 272.9 and ICD-10-CM codes E78.0, E78.1, E78.2, E78.3, E78.4, and E78.5), coronary artery disease (CAD, (ICD-9-CM codes 410, 411, 412, 413, and 414 and ICD-10-CM codes I20, I21, I22, I24, and I25)), and congestive heart failure (CHF, (ICD-9-CM codes 402.01, 402.11, 402.91, 404.01, 404.03, 404.11, 404.13, 404.91, 404.93, and 428 and ICD-10-CM codes I11.0, I13.0, I13.2, and I50)). The claims-based stroke severity index (SSI) was employed to assess stroke severity. Next, the SSI was converted to the National Institutes of Health Stroke Scale score by using the following equation: estimated National Institutes of Health Stroke Scale (eNIHSS) = 1.1722 × SSI − 0.7533 [15]. Measurements of body mass index (BMI), systolic blood pressure (SBP), diastolic blood pressure (DBP), heart rate (HR), creatinine, alanine aminotransferase, and lipid profiles were obtained from the records of the enrolled patients. HR, SBP, and DBP were measured with patients lying down after 5 min of rest and were recorded using an automated oscillometric device (DINAMAP ProCare 100, GE Medical Systems, Milwaukee, WI, USA) or a bedside patient monitor (IntelliVue MP60, Philips Medical System, Boeblingen, Germany). If the pulse was irregular, the HR was measured by palpating the radial pulse over 60 s. The mean SBP, DBP, and HR, along with the HR-CV, were derived from the vital sign data recorded over the first 3 days of hospitalization. We used the following formulae to calculate the HR-CV and standard deviation in heart rate (SD): HR-CV =$\;\left(\frac{SD}{\bar{HR}}\right)$; SD =$\sqrt{\frac{\sum {\left(HRi-\bar{HR}\right)}^{2}}{n-1}}$, where HRi is the HR of the patient at the examination and $\bar{\mathrm{HR}}$ is the mean HR. The coefficient of variation was selected as a measure of variation because it is more independent of the mean than the standard deviation.

### 2.4. Statistical Analysis

Descriptive statistics are presented as numbers (percentages) for categorical variables. For continuous variables, they are presented as means (standard deviations) and medians (interquartile ranges). Between-group differences in continuous and categorical variables were tested using the Kruskal–Wallis rank sum test and the chi-square test, respectively. Multivariable logistic regression analysis was conducted to evaluate the association between HR-CV and paroxysmal AF as well as the association between HR-CV and persistent AF. Formal analyses were performed with HR-CV as both a continuous and categorical variable (patients were stratified into four subgroups according to HR-CV levels, namely HR-CV < 0.08, HR-CV ≥ 0.08 and <0.10, HR-CV ≥ 0.10 and <0.12, and HR-CV ≥ 0.12). The associations of HR-CV with subsequent paroxysmal or persistent AF were estimated using logistic regression models. In addition to crude odds ratios (ORs), adjusted ORs and 95% confidence intervals (CIs) were calculated with reference to the lowest risk group and were estimated after adjustment for potential confounders in the logistic regression models. Potential confounders in model 1 comprised age, sex, and eNIHSS scores, and potential confounders in model 2 comprised age, sex, eNIHSS scores, hypertension, diabetes mellitus, dyslipidemia, CAD, CHF, history of cancer, smoking status, BMI, total cholesterol, triglycerides, creatinine, alanine aminotransferase, mean HR, mean SBP, and mean DBP. Unless specified, all results are given for the fully adjusted model. C-statistics were calculated to assess the ability of the multivariable model to predict paroxysmal and persistent AF. The restricted cubic spline smoothing technique was employed to explore the overall trend of associations through the range of HR-CV values.

Subgroup analyses were performed with HR-CV as a continuous variable. ORs and 95% CIs for each subgroup were calculated for every increment of 1 standard deviation in HR-CV. All analyses were performed using IBM SPSS Statistics for Windows, version 22 (IBM Corp., Armonk, NY, USA) and R software, version 4.0.0 (R Foundation for Statistical Computing, Vienna, Austria).

This study was conducted according to the tenets of the Declaration of Helsinki. All procedures were conducted in adherence to relevant guidelines and regulations.

## 3. Results

### 3.1. Baseline Characteristics

The analysis included 8179 patients with AIS (mean age, 68.05 ± 13.56 years; 61.1% men). The mean SBP and DBP were 148.36 ± 19.35 and 83.85 ± 11.26 mmHg, respectively, and the mean HR was 75.84 ± 12.01 beats per minute (bpm). The number of HR measurements totaled 194,715, with the median number of measurements per patient being 12 (interquartile range: 9–20). The clinical characteristics of the included patients are presented in Table 1. Based on the baseline 12-lead ECG results, 2892 patients and 18,210 patients received and did not receive a diagnosis of AF, respectively. Among patients without AF on 12-lead ECG, 5287 patients underwent further 24 h Holter monitoring, and 274 patients were diagnosed as having paroxysmal AF. In total, 5013 patients without AF, 2725 patients with persistent AF, and 441 patients with paroxysmal AF were included in the analysis (Figure 1). Overall, 5.4% and 33.3% of patients had paroxysmal and persistent AF, respectively. Compared with patients without AF, those with AF were more likely to be older; female; not current smokers; to have CAD and CHF; to have higher eNIHSS scores and a higher mean HR; to have lower mean SPB and DBP; to have lower BMI; lower frequency of diabetes mellitus and dyslipidemia; higher baseline creatinine levels; and lower baseline total cholesterol, triglyceride, and alanine aminotransferase levels.

The demographic data and baseline characteristics of patients for each HR-CV subgroup are displayed in Appendix A. Compared with patients with a lower HR-CV, those with a higher HR-CV were more likely to be older, not to be current smokers, not to have diabetes mellitus and dyslipidemia, and to have higher eNIHSS scores, higher frequency of CHF, and lower baseline total cholesterol and triglyceride levels.

### 3.2. Association of HR-CV Level with AF

The detection rates of patients without AF, with paroxysmal AF, and with persistent AF according to different HR-CV levels in the study population are shown in Appendix A. The effects of every increment of one standard deviation in HR-CV on the associations of HR-CV with paroxysmal and persistent AF are summarized in Table 2. In addition to high HR-CV levels, mean HR, older age, being female, eNIHSS scores, and CAD were all positively associated with paroxysmal AF. Conversely, a history of dyslipidemia and higher mean SBP levels were negatively associated with paroxysmal AF. In addition to higher HR-CV levels, older age, being female, eNIHSS scores, BMI, mean HR levels, mean DBP levels, and CHF were positively associated with persistent AF. A history of dyslipidemia, total cholesterol, triglyceride, and mean SBP levels were negatively associated with persistent AF.

Crude and adjusted ORs of paroxysmal and persistent AF for each HR-CV subgroup are given in Figure 2. Compared with the reference group (the HR-CV < 0.08 subgroup), the adjusted ORs of paroxysmal AF in model 2 were 1.866 (95% CI = 1.205–2.889) for the HR-CV ≥ 0.08 and <0.10 subgroup, 1.889 (95% CI = 1.174–3.038) for the HR-CV ≥ 0.10 and <0.12 subgroups, and 5.564 (95% CI = 3.847–8.047) for the HR-CV ≥ 0.12 subgroup. Compared with the HR-CV < 0.08 subgroup, the adjusted ORs of persistent AF in model 2 were 2.425 (95% CI = 1.921–3.062) for the HR-CV ≥ 0.08 and <0.10 subgroup, 4.312 (95% CI = 3.415–5.446) for the HR-CV ≥ 0.10 and <0.12 subgroup, and 5.651 (95% CI = 4.586–6.964) for the HR-CV ≥ 0.12 subgroup (Figure 2). The C-statistics of paroxysmal and persistent AF were 0.827 and 0.862, respectively.

Even after multiple adjustments for potential confounders, no U-shaped curves were found for the associations of HR-CV with paroxysmal and persistent AF. Higher HR-CV levels were significantly and persistently associated with elevated ORs of paroxysmal and persistent AF (Figure 3).

### 3.3. Subgroup Analysis

The results of subgroup analyses are presented in Figure 4. Associations of HR-CV with paroxysmal and persistent AF were found in all the subgroups. Moreover, a substantial impact of elevated HR-CV levels on OR of paroxysmal AF was observed when patients had a mean HR of ≥90 bpm.

### 3.4. Interaction of Mean HR and HR-CV

To test for an interaction between mean HR and HR-CV on ORs of AF, patients were divided into four subgroups according to their HR-CV (<0.12 or ≥0.12) and mean HR (<90 or ≥90 bpm). The associations with paroxysmal and persistent AF were greatest in patients with an HR-CV ≥ 0.12 and a mean HR ≥ 90 bpm as compared with patients with an HR-CV of <0.12 and a mean HR of <90 bpm (OR = 6.316, 95% CI = 3.824–10.434 for paroxysmal AF and OR = 4.333, 95% CI = 3.073–6.108 for persistent AF) (Appendix A).

## 4. Discussion

In this study, HR-CV was highly associated with both paroxysmal and persistent AF, independent of other known associated factors such as age and stroke severity (Table 2).

The size of the study population allowed us to adjust for two of the strongest AF-associated factors in the multivariable model: age and stroke severity [16]. In the subgroup analysis of patients with different age ranges and varying stroke severity, the associations of HR-CV with paroxysmal and permanent AF were consistent (Figure 4).

In the ONTARGET/TRANSCEND studies, low mean HR (<60 bpm) was independently associated with incident AF, and low visit-to-visit HRV and high SBP further increased the incidence of new-onset AF in patients at high risk of cardiovascular disease [17]. In a large community-based cohort study, visit-to-visit HRV was linked to an elevated risk of new-onset AF with a nonlinear U-shaped association [8]. The objective of these studies was to predict new-onset AF during long-term follow-up, whereas we assessed the risk of AF during diagnostic workup in stroke units. Considering between-study differences in study objectives, the effects of mean HR and HR-CV on the risk of AF may vary. Herein, the associations of HR-CV with paroxysmal and persistent AF progressively increased in strength as HR-CV increased. No clear evidence of U-shaped associations of HR-CV with paroxysmal and persistent AF was noted (Figure 3).

Because AF is characterized by a rapid and irregular heart rhythm, the mean HR and visit-to-visit HRV of patients with AIS during hospitalization are expected to be positively associated with AF. In this study, a synergistic interaction was observed between mean HR and HR-CV. High HR-CV was independently associated with paroxysmal AF, and high mean HR further increased the risk of paroxysmal AF in patients with AIS (Figure 4). The association of HR-CV with AF was stronger when both mean HR and HR-CV were higher. Compared with those of the reference group (mean HR < 90 bpm and HR-CV < 0.12), the ORs of paroxysmal and persistent AF were significantly higher in patients with a mean HR of ≥ 90 bpm and an HR-CV of ≥0.12 (OR = 6.316, 95% CI = 3.824–10.434 for paroxysmal AF; and OR = 4.333, 95% CI = 3.073–6.108 for persistent AF; Appendix A).

Visit-to-visit HRV has been reported to be associated with various cardiovascular diseases [7,18,19]. The correlation between beat-to-beat HRV and AF is due to cardiac autonomic dysfunction [20]. Although the exact mechanism through which visit-to-visit HRV is associated with AF risk remains unclear, in this study, patients with higher HR-CV levels tended to be older, have higher stroke severity, have lower levels of total cholesterol and triglycerides, and have a higher frequency of CHF and lower frequency of dyslipidemia than those with lower HR-CV levels (Appendix A). All these factors have been noted to be highly associated with AF [16,21,22,23,24].

In the multivariable logistic regression analysis, age, sex, history of dyslipidemia, mean SBP, and mean HR were associated with paroxysmal and persistent AF (Table 2). As expected, age was a strong predictor of AF [21,25]. Although the lifetime risk of AF was higher in men than in women [26], women with AF are at a higher overall risk of thromboembolic stroke than men with AF [27]. Consistent with the finding of another study [28], women with stroke were more likely to have AF than their male counterparts in the current study. Moreover, dyslipidemia and mean SBP were negatively associated with paroxysmal and persistent AF. Several large community-based cohort studies have detected inverse associations between blood lipid levels and AF incidence [24,29,30,31], and these observations are compatible with the present results (Table 1 and Table 2). Although hypertension was previously identified as a risk factor for the development of AF [32], we observed that after AIS, patients with paroxysmal and persistent AF tended to have lower mean SBP levels, as measured at the start of the hospitalization period, than did those without AF (Table 1); this observation is in line with the findings of other studies [33,34,35]. Lower SBP levels during the acute stage in patients with AIS and AF may be correlated with the mechanism of the stroke subtype rather than with baseline blood pressure levels. Therefore, dyslipidemia and mean SBP, as measured at the start of the hospitalization period, were negatively associated with AF.

Although AF is frequently associated with cardiac disease, including CAD and CHF, the frequency of cardiac disease usually differs in individuals with paroxysmal and persistent AF [23,36,37]. Herein, due to the distinct frequency of CAD and CHF in AIS patients with paroxysmal AF, persistent AF, and without AF, CAD and CHF were associated with paroxysmal and persistent AF in the multivariable logistic regression analysis, respectively (Table 2).

To provide stroke patients with AF with the appropriate anticoagulant therapy, the timely detection of AF is warranted. As shown in Figure 3, the relationship between HR-CV and ORs of persistent AF was relatively linear, whereas the relationship between HR-CV and ORs of paroxysmal AF was exponential. Therefore, HR-CV, as derived from HR measurements taken in the first 3 days of hospitalization, constitutes a useful parameter for identifying paroxysmal AF in patients with AIS. The ORs of paroxysmal AF for the HR-CV ≥ 0.12 subgroup were substantially greater than those for the HR-CV < 0.08 subgroup (OR = 5.564, 95% CI 3.847–8.047; Figure 2). AIS patients with an HR-CV of ≥0.12 should be considered at high risk of paroxysmal AF and should undergo long-term cardiac monitoring during diagnostic workup in stroke units.

## 5. Limitations

This study has several limitations. First, to collect vital sign data over the first 3 days of hospitalization, patients hospitalized for fewer than 3 days were not included in this study. Therefore, some patients with mild or extremely severe stroke may have been excluded. Second, given the focus on patients with AIS, the findings may not apply to patients with other cardiovascular diseases. Third, HR-CV was determined using HR data collected over the first 3 days of hospitalization. However, we did not record HR values in the same intervals during hospitalization; this may have led to the underestimation or overestimation of the association between HR-CV and AF. Fourth, the absence of paroxysmal AF was determined through 24 h Holter monitoring. The use of this procedure alone may not be sufficient to detect all instances of paroxysmal AF, resulting in underdiagnosis. Finally, we did not collect data on pharmacotherapy, brain-imaging study, and echocardiographic information on cardiac structure for the study patients as potential confounders in the analysis.

## 6. Conclusions

Visit-to-visit HRV expressed as HR-CV was significantly associated with the risk of AF—specifically both paroxysmal and persistent AF. HR-CV can be easily derived from routine vital sign measurements. High HR-CV can be employed to identify patients with AIS who are at high risk of paroxysmal AF.

## Figures and Tables

**Figure 1 jcm-12-01050-f001:**
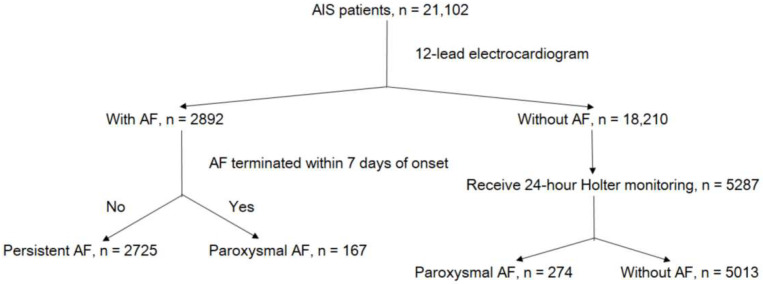
Flowchart of patient selection. Abbreviations: AIS, acute ischemic stroke; AF, atrial fibrillation.

**Figure 2 jcm-12-01050-f002:**
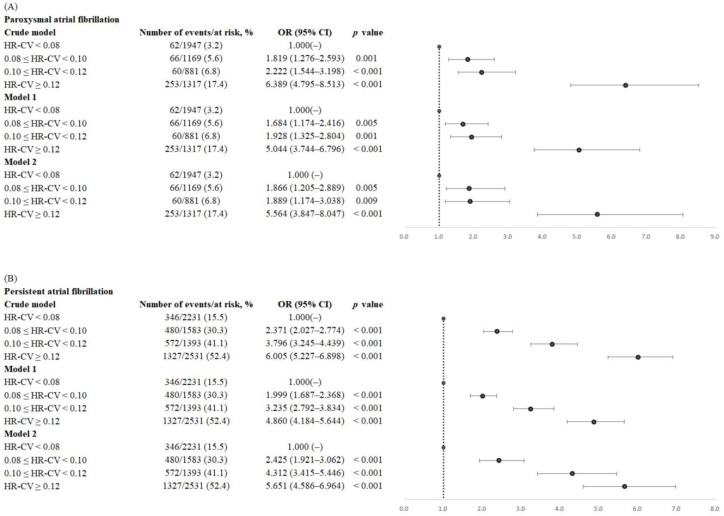
Forest plots of crude and adjusted ORs (95% CIs) of (**A**) paroxysmal atrial fibrillation and (**B**) persistent atrial fibrillation by HR-CV increments. The analyses were adjusted for age, sex, and eNIHSS score in model 1, as well as for all confounders considered in the fully adjusted model (model 2), namely age, sex, eNIHSS score, history of hypertension, diabetes mellitus, dyslipidemia, coronary artery disease, congestive heart failure, a history of cancer, smoking status, body mass index, total cholesterol, triglycerides, creatinine, alanine aminotransferase, mean heart rate, mean systolic blood pressure, and mean diastolic blood pressure. Abbreviations: OR, odds ratio; CI, confidence interval; HR-CV, coefficient of variation in HR; eNIHSS, estimated National Institutes of Health Scale.

**Figure 3 jcm-12-01050-f003:**
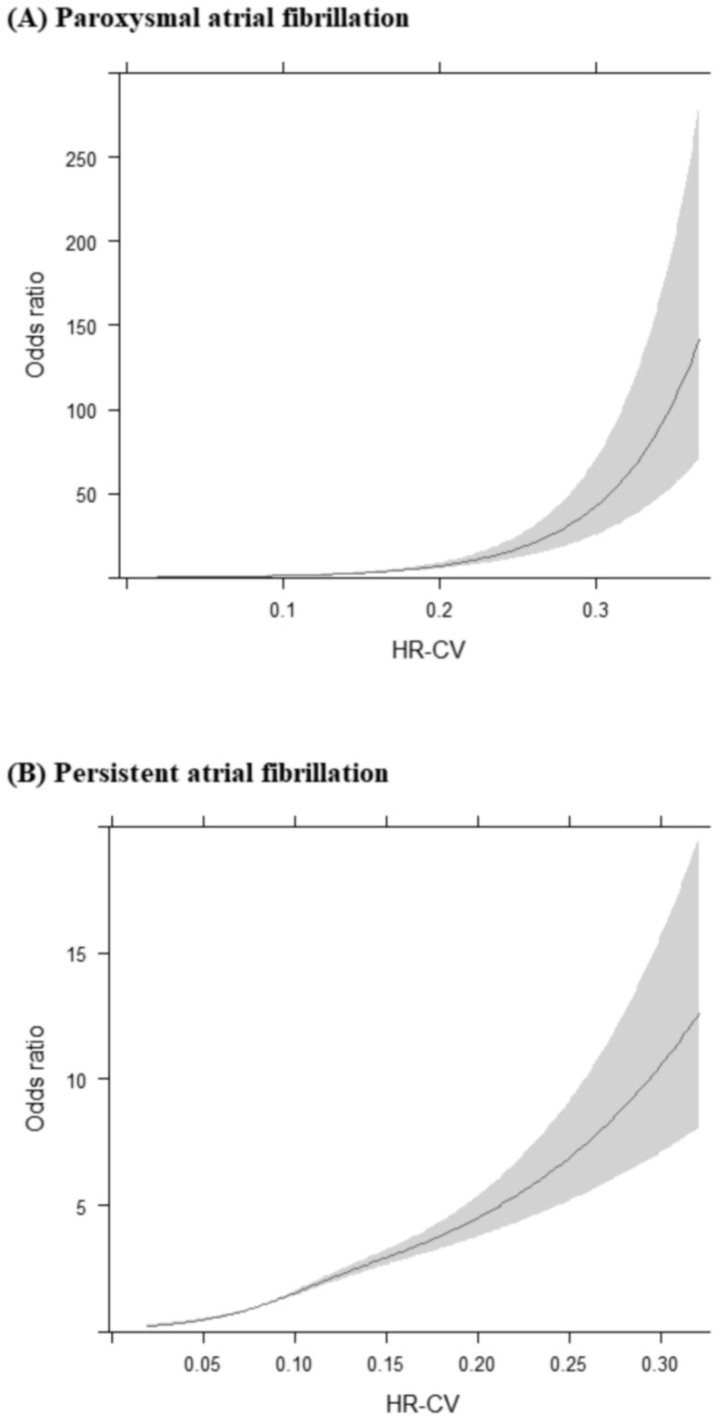
Restricted cubic splines are presented for the associations between HR-CV levels and (**A**) paroxysmal atrial fibrillation as well as (**B**) persistent atrial fibrillation. The analyses were adjusted for all confounders considered in the fully adjusted model, namely age, sex, eNIHSS score, history of hypertension, diabetes mellitus, dyslipidemia, coronary artery disease, congestive heart failure, history of cancer, smoking status, body mass index, total cholesterol, triglycerides, creatinine, alanine aminotransferase, mean heart rate, mean systolic blood pressure, and mean diastolic blood pressure. Abbreviations: HR-CV, coefficient of variation in heart rate; eNIHSS, estimated National Institutes of Health Scale.

**Figure 4 jcm-12-01050-f004:**
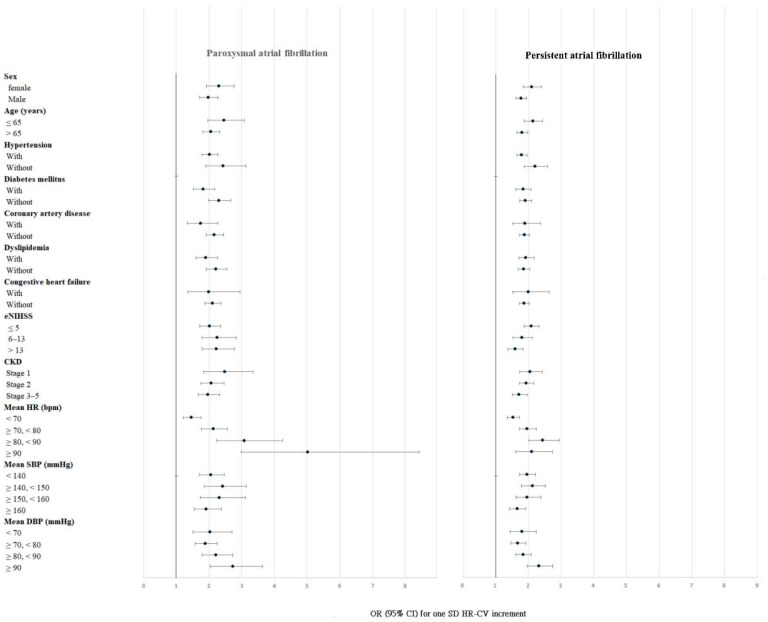
Subgroup analyses of paroxysmal and persistent atrial fibrillation per standard deviation of HR-CV increments. Abbreviations: HR-CV, coefficient of variation in heart rate; eNIHSS, estimated National Institutes of Health Stroke Scale; CKD, chronic kidney disease; HR, heart rate; SBP, systolic blood pressure; DBP, diastolic blood pressure; OR, odds ratio; CI, confidence interval; SD, standard deviation.

**Table 1 jcm-12-01050-t001:** Baseline characteristics of the study population.

	Acute Ischemic Stroke				*p* Value
	Total (N = 8179)	Without AF (N = 5013)	With Paroxysmal AF (N = 441)	With Persistent AF (N = 2725)	
Age, years					<0.001
Mean (SD)	68.05 (13.56)	64.30 (13.67)	74.73 (10.38)	73.86 (11.15)	
Median (Q1, Q3)	69.00 (59.00, 78.00)	65.00 (55.00, 75.00)	76.00 (68.00, 82.00)	75.00 (66.00, 82.00)	
Male	4997 (61.1)	3340 (66.6)	240 (54.4)	1417 (52.0)	<0.001
eNIHSS					<0.001
Mean (SD)	8.34 (6.64)	6.58 (4.96)	9.78 (7.26)	11.34 (7.98)	
Median (Q1, Q3)	4.06 (4.06, 10.90)	4.06 (4.06, 6.00)	5.66 (4.06, 16.27)	8.95 (4.06, 20.04)	
Hypertension	5931 (72.5)	3662 (73.1)	336 (76.2)	1933 (70.9)	0.028
Diabetes mellitus	2837 (34.7)	1874 (37.4)	129 (29.3)	834 (30.6)	<0.001
Dyslipidemia	3600 (44.0)	2592 (51.7)	139 (31.5)	869 (31.9)	<0.001
Congestive heart failure	687 (8.4)	212 (4.2)	55 (12.5)	420 (15.4)	<0.001
Coronary artery disease	871 (10.6)	445 (8.9)	79 (17.9)	347 (12.7)	<0.001
Current smoker	2180 (26.7)	1630 (32.5)	97 (22.0)	453 (16.6)	<0.001
History of cancer	478 (5.8)	279 (5.6)	29 (6.6)	170 (6.2)	0.385
Body mass index, kg/m^2^					<0.001
Mean (SD)	24.76 (4.26)	25.05 (4.32)	24.01 (3.96)	24.35 (4.17)	
Median (Q1, Q3)	24.40 (21.91, 27.12)	24.66 (22.20, 27.34)	23.92 (21.23, 26.67)	24.07 (21.45, 26.69)	
Total cholesterol, mmol/L					<0.001
Mean (SD)	4.55 (1.09)	4.69 (1.13)	4.36 (1.06)	4.32 (0.99)	
Median (Q1, Q3)	4.45 (3.83, 5.15)	4.58 (3.96, 5.31)	4.27 (3.66, 4.91)	4.25 (3.68, 4.90)	
Triglycerides, mmol/L					<0.001
Mean (SD)	1.41 (1.05)	1.56 (1.08)	1.26 (0.85)	1.15 (0.97)	
Median (Q1, Q3)	1.18 (0.84, 1.67)	1.31 (0.94, 1.86)	1.05 (0.79, 1.46)	0.97 (0.72, 1.33)	
Creatinine, μmol/L					<0.001
Mean (SD)	112.09 (115.82)	111.25 (120.01)	126.77 (138.48)	111.26 (103.13)	
Median (Q1, Q3)	85.75 (69.84, 109.62)	83.98 (68.95, 106.96)	91.94 (73.37, 118.90)	88.40 (70.72, 114.04)	
Alanine aminotransferase, U/L					<0.001
Mean (SD)	26.42 (38.14)	26.86 (25.90)	23.78 (19.01)	26.05 (55.33)	
Median (Q1, Q3)	21.00 (16.00, 29.00)	21.00 (16.00, 29.00)	20.00 (14.00, 27.00)	20.00 (15.00, 28.00)	
Mean SBP, mmHg					<0.001
Mean (SD)	148.36 (19.35)	150.51 (19.73)	145.29 (18.78)	144.90 (18.15)	
Median (Q1, Q3)	147.20 (134.50, 161.60)	149.43 (136.39, 164.06)	144.50 (132.63, 156.78)	144.01 (132.12, 157.24)	
Mean DBP, mmHg					<0.001
Mean (SD)	83.85 (11.26)	84.76 (11.17)	79.49 (11.46)	82.90 (11.15)	
Median (Q1, Q3)	83.14 (76.37, 90.91)	84.00 (77.14, 91.76)	79.00 (72.61, 86.06)	82.30 (75.44, 90.19)	
Mean heart rate, bpm					<0.001
Mean (SD)	75.84 (12.01)	73.19 (10.71)	77.68 (13.41)	80.42 (12.59)	
Median (Q1, Q3)	74.86 (67.62, 82.83)	72.74 (65.92, 79.36)	75.89 (68.76, 85.46)	79.88 (12.59, 89.19)	
CV in heart rate					<0.001
Mean (SD)	0.109 (0.048)	0.097 (0.039)	0.148 (0.078)	0.126 (0.047)	
Median (Q1, Q3)	0.101 (0.077, 0.133)	0.090 (0.069, 0.118)	0.132 (0.095, 0.181)	0.118 (0.093, 0.149)	

Abbreviations: AF, atrial fibrillation; SD, standard deviation; Q, quartile; eNIHSS, estimated National Institute of Health Stroke Scale; SBP, systolic blood pressure; DBP, diastolic blood pressure; bpm, beats per minute; CV, coefficient of variation.

**Table 2 jcm-12-01050-t002:** Multivariable logistic regression analysis including the HR-CV as a continuous variable for paroxysmal and persistent atrial fibrillation.

	Paroxysmal Atrial Fibrillation		Persistent Atrial Fibrillation	
	OR (95% CI)	*p* Value	OR (95% CI)	*p* Value
HR-CV (per 1 SD)	2.082 (1.867–2.322)	<0.001	1.887 (1.748–2.036)	<0.001
Male	0.652 (0.484–0.879)	0.005	0.612 (0.519–0.721)	<0.001
Age	1.057 (1.044–1.071)	<0.001	1.067 (1.060–1.075)	<0.001
Hypertension	1.048 (0.759–1.446)	0.776	0.965 (0.811–1.149)	0.693
Diabetes mellitus	0.838 (0.628–1.117)	0.228	0.997 (0.852–1.167)	0.972
Current smoker	1.039 (0.736–1.467)	0.829	0.835 (0.694–1.004)	0.055
Coronary artery disease	1.823 (1.260–2.635)	0.001	1.128 (0.895–1.421)	0.308
Dyslipidemia	0.559 (0.420–0.744)	<0.001	0.630 (0.540–0.735)	<0.001
Congestive heart failure	1.543 (0.975–2.443)	0.064	2.515 (1.938–3.265)	<0.001
History of cancer	0.717 (0.408–1.261)	0.248	0.928 (0.688–1.253)	0.627
eNIHSS	1.015 (0.994–1.037)	0.168	1.074 (1.061–1.086)	<0.001
Body mass index	1.017 (0.983–1.052)	0.326	1.042 (1.024–1.061)	<0.001
Total Cholesterol	0.985 (0.864–1.123)	0.822	0.899 (0.832–0.970)	0.006
Triglycerides	0.974 (0.823–1.152)	0.757	0.756 (0.678–0.844)	<0.001
Alanine aminotransferase	0.998 (0.993–1.004)	0.600	1.000 (0.999–1.002)	0.719
Creatinine	1.001 (1.000–1.002)	0.216	1.000 (0.999–1.001)	0.816
Mean heart rate	1.020 (1.008–1.033)	0.002	1.027 (1.020–1.035)	<0.001
Mean systolic blood pressure	0.986 (0.977–0.996)	0.006	0.961 (0.955–0.966)	<0.001
Mean diastolic blood pressure	0.995 (0.977–1.013)	0.587	1.065 (1.054–1.076)	<0.001

Abbreviations: HR-CV, coefficient of variation in heart rate; OR, odds ratio; CI, confidence interval; SD, standard deviation; eNIHSS, estimated National Institute of Health Stroke Scale.

## Data Availability

The data that support the findings of this study are available from Chang Gung Research Databank at Chang Gung Memorial Hospital, Chiayi Branch, Taiwan, but restrictions apply to the availability of these data, which were used under license for the current study, and so are not publicly available. Data are, however, available from the authors upon reasonable request and with permission of the local Institutional Review Board of Chang Gung Memorial Hospital, Chiayi Branch, Taiwan.

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
