# Peer review of "Initial In-Hospital Visit-to-Visit Heart Rate Variability Is Associated with Higher Risk of Atrial Fibrillation in Patients with Acute Ischemic Stroke"

_jcm, 2023, doi:10.3390/jcm12031050_

Round 1

Reviewer 1 Report

Lee et al has performed an interesting study assessing the implication of HR variability in new onset AF prediction among patients with acute ischemic stroke. The authors have found that the higher the HRV, the more positive the correlation with new onset AF. The authors should be commended on this exhaustive analysis which in the future would confer significant clinical implication such that this condition can be better cared. Nonetheless, there are some perspectives the writer would like to query

1. How do you define HR-CV? Can the authors provide the formula or algorithm to  calculate such measure?

2. Are there any echocardiogram findings which can be used for analysis? For example, left atrial size, EF, PASP, e/e' etc.

3. Any medications which can be used to adjusted? Essentially statin, as low cholesterol on those patients may be low due to the medications intake.

4. I suggest other comorbidities including DM2, CAD should also be added. In one study, these 2 factors can be highly associated with risk of AF 

5. To clarify, for the endpoint, new onset AF, did the author follow up only 3 days when patients were hospitalized or extend to outpatient follow up? Please clarify

Reviewer 2 Report

Review of manuscript JCM Nr.2073372

Dear authors,

I have read with interest your article. It is a interesting topic, since patients with ischemic stroke are at higher risk of developing atrial fibrillation (afib). Moreover paroxysmal atrial fibrillation is very often the cause of the originaly cryptogenic  stroke (embolic stroke of unknown origin, ESUS). The trouble is that you have to catch the Afib on ECG (even from samrtwatch i tis sufficient) in order to start the therapy. If you only assess the risk level of developing Afib, you cannot start any anticoagulation treatment in order to prevent the stroke. In my opinion this is the major limitation of your study. 

The othe significant limitations is peforming HRV analyses in persistent atrial fibrillation, that is nonsense – explained further bellow. Of note, you are prediciting Afib occurrence in Afib, which already accured in case of persistent Afib (called sustained in your paper).

From the formal point of view the paper is well written with adequate statistics and formidable graphics. 

There are also some other issues to be solved., namely: 

-       In the age of smartwatch and similar technology i do not find the diagnosis of Afib so much resource-intensive

-       Please use standard terminology for atrial fibrillation;thus not sustained but persistent or long standing persistent according to AHA/ESC guidelines for management of atrial fibrillation

-       Why only HRV-CV was tested, there are plenty of other HRV measures /time-domain analysis (SDNN, SDANN indexes etc.), spectral analysis or non-linear methods (aproximate entropy/detrended fluctuation), heart rate turbulence/

-       Have you xecluded the possibility of premature atrial contractions (PAC) accounting for difference in the HRV-CV measure. Since PAC’s are well known for prediciting Afib occurence, moreover in case of excessive PAC, you may insitute anticoagulation treatment according to the subclinical atrial fibrillation guide (ESC/EHRA)

-       Performing HRV-CV in persistent atrial fibrillation is from my viewpoint almost misconduct sice the very nature of Afib is irregularity. Definitely a waste of time. It does not have any predicitve value. HRV is meaningull only in SR, since Afib irregularity does not dependet on autonomus systém variation only but i tis naturally irregular due to left atrial activation patterns and Afib triggers!

Round 2

Reviewer 1 Report

Authors' response have mostly covered my queries. 

Reviewer 2 Report

Dear authors,

we are still not clear about the persistent AF. If the patient you categorized as having AF and it did persist more than 7 days, how you could have measured the HRV? If the measurement should have any meaning, you have to measure during sinus rhythm, not AF.

This issue has to be solved! If the measurement was performed during atrial fibrillation, it has almost no meaning at all. How could you predict AF with AF-based HRV? You can't!

May I got it wrong, but than you have to explained it bette in the methods. Also persistent AF should be defined in the appropriate section. You defined paroxysmal, but not persistent. For persistent AF definition you have to hopitalizaze the patients at least 7-days or have 7-day ECG - holler. 

Lets say, you measured all the HRV in sinus rhythm (what is at a time not clear). Why the HRV is predictive only of paroxysmal AF, when according to your results the persistent AF has higher odds ratio. 

In figure 1 you have mentioned both persistent and sustained AF, that is a minor issues, but is definitely confusing. 
